# The Optimized Workflow for Sample Preparation in LC-MS/MS-Based Urine Proteomics

**DOI:** 10.3390/mps2020046

**Published:** 2019-06-07

**Authors:** Suguru Saito, Yoshitoshi Hirao, Ali F. Quadery, Bo Xu, Amr Elguoshy, Hidehiko Fujinaka, Shohei Koma, Keiko Yamamoto, Tadashi Yamamoto

**Affiliations:** 1Biofluid Biomarker Center, Niigata University, Niigata 9502181, Japan; s.saito.csmc@gmail.com (S.S.); hirao@ccr.niigata-u.ac.jp (Y.H.); fquadery@yahoo.com (A.F.Q.); kyonami-bbc@ccr.niigata-u.ac.jp (B.X.); amr_biotech2006@yahoo.com (A.E.); dext007@hotmail.com (H.F.); yamamotok-bbc@ccr.niigata-u.ac.jp (K.Y.); 2Division of Clinical Nephrology and Rheumatology, Kidney Research Center, Niigata University Graduate School of Medical and Dental Sciences, Niigata 9518510, Japan; 3Department of Pediatrics, National Hospital Organization Niigata Hospital, Kashiwazaki 9458585, Japan; 4Life Science Research Laboratory, Tosoh Corporation, Kanagawa 252-1123, Japan; koma@tosoh-arc.co.jp; 5Department of Clinical laboratory, Shinrakuen Hospital, Niigata 9502087, Japan

**Keywords:** Urine proteomics, Urine protein precipitation, Urine biomarker

## Abstract

The sample condition is an important factor in urine proteomics with stability and accuracy. However, a general protocol of urine protein preparation in mass spectrometry analysis has not yet been established. Here, we proposed a workflow for optimized sample preparation based on methanol/chloroform (M/C) precipitation and in-solution trypsin digestion in LC-MS/MS-based urine proteomics. The urine proteins prepared by M/C precipitation showed around 80% of the protein recovery rate. The samples showed the largest number of identified proteins, which were over 1000 on average compared with other precipitation methods in LC-MS/MS-based urine proteomics. For further improvement of the workflow, the essences were arranged in protein dissolving and trypsin digestion step for the extraction of urine proteins. Addition of Ethylene diamine tetraacetic acid (EDTA) dramatically enhanced the dissolution of protein and promoted the trypsin activity in the digestion step because the treatment increased the number of identified proteins with less missed cleavage sites. Eventually, an optimized workflow was established by a well-organized strategy for daily use in the LC-MS/MS-based urine proteomics. The workflow will be of great help for several aims based on urine proteomics approaches, such as diagnosis and biomarker discovery.

## 1. Introduction

The effectiveness of urine proteomics has been recognized year by year because urine reflects various biological phenomenon in our body [1]. In addition, urine is collected in a non-invasive way, so it can be used for various examinations and clinical tests frequently. Urine examination has some benefits to detect physiological changes and diseases in an individual. Blood examination also shows internal environment clearly; however, it has a problem that the sampling itself is invasive and is difficult to repeat several times during the short term [2]. Many studies conducted examinations utilizing urine samples, and those results provided valuable information in various medical and health care fields [3]. Recently, urine proteomics is actively performed to figure out the internal environment change in the human body. Especially, an investigation of biomarkers for various diseases is intensively performed using urine proteomics approach [4]. Throughout some studies, several biomarkers have already been discovered for the detection and prediction of diseases [5,6].

The most important factors in urine proteomics are the quality and size of LC-MS/MS analysis data. These factors are influenced by not only mass spectrometry and software used in the analysis but also by the urine protein sample condition itself. A standard guide was proposed in urine sample collection and storage for proteomics by Human Kidney and Urine Proteome Project group of Human Proteome Organization (HKUPP) [7]; however, the general protocol has not yet been established in urine protein preparation for LC-MS/MS analysis. The urine proteins for proteomics are generally extracted from fresh or frozen urine by various methods depending on the aim of the study. For instance, the samples were prepared by organic solvent high-speed centrifugation, fractionation, and using commercial kit [8,9,10]. Even though all the methods seemed to be available for LC-MS/MS-based urine proteomics, those samples showed different conditions, especially in the proportion and the purity of extracted proteins. The differences might have been reflected in the final data of the identified proteins, which means the total number and specificity were unstable between each analysis.

Here, we showed an optimized workflow in urine proteomics. The workflow firstly focused on the method of urine protein extraction by organic solvent high-speed centrifugation, especially from frozen urine, then peptide preparation by in-solution trypsin digestion was considered as well. Since the homogeneity and recovery rate of extracted urine protein are indispensable in LC-MS/MS-based quantitative proteomics, several precipitation methods for urine protein preparation (ethanol, acetone, acetonitrile, and methanol/chloroform (M/C)) were compared. The extracted protein recovery rate in ethanol and M/C precipitation reached around 80% on average, which was better than other methods. The samples achieved over 1000 identified proteins, and M/C precipitation showed the highest percentage of unique proteins (9.9%) in the analysis. We focused on the purity of the precipitated proteins as well because urine contained a large amount of staining substances, such as urobilin, which cannot be eliminated easily once it is precipitated together with urine proteins. We found an effective treatment in which Tris-HCl addition into urine eliminated the contaminants as much as possible from precipitated urine proteins. Furthermore, the combination of M/C precipitation and Tris-HCl addition showed the best quality of the precipitated urine proteins. To improve the efficiency of trypsin digestion, we focused on the dissolution of extracted urine proteins. The urine proteins extracted by M/C precipitation were strongly dehydrated, so the protein pellet had poor solubility in 8 M urea/50 mM Tris-HCl (pH 8.0), which is generally used in protein dissolving [11]. We achieved a great improvement in the solubility using Ethylene Diamine Tetra Acetic acid (EDTA)-containing buffer. The sample treated with the buffer showed an increase in protein identification (+5.2%) compared with other samples treated with different buffers. Furthermore, EDTA buffer contributed to reducing missed cleavages.

Throughout this study, we proposed an optimized workflow mainly for sample preparation step in urine proteomics. The workflow will contribute to promoting urine protein quality used for LC-MS/MS-based urine proteomics.

## 2. Materials and Methods

### 2.1. Human Urine Sample and Protein Preparation

Urine samples were collected from healthy volunteers (age 20–40, male) in 50 mL tube. To remove insoluble materials and cellular debris, the samples were centrifuged at 1000× *g* for 10 min. The supernatants were harvested and separated as aliquots in 1.5 mL tubes and stored at −20 °C until use. The frozen urine was used for analysis within 6 months. These frozen samples were thawed in a water bath at 37 °C for 10 min before use. Once the frozen urine was thawed, it was all used for analysis to prevent freeze and thaw cycles. The institutional ethics committees at Niigata University and corona corporation approved this study.

### 2.2. Study Design for Characterization and Evaluation of Each Precipitation Method

This study used the frozen urine sample described in Section 2.1. The urine was separated into three different tubes (250 µL in each) from the original stock. Two tubes were used for urine protein precipitation. The third tube was used for the original urine protein assay. One of the precipitated samples was used for trypsin digestion and peptide purification. Then, it was analyzed by LC-MS/MS, and the data was used for protein identification. The other sample was used for precipitated protein recovery check by protein assay, according to the formula described in Section 2.4.

### 2.3. Precipitation Methods for Urine Protein Preparation

#### 2.3.1. Ethanol Precipitation

The five times volume (1250 µL) of 100% ethanol was added to 250 µL of urine and mixed well for 5 min. The sample was kept at −20 °C for overnight. The sample was centrifuged at 12,000× *g* for 15 min. The supernatant was removed, then the pellet was washed with 1000 μL of 70% ethanol. The sample was again centrifuged at 12,000× *g* for 5 min. The supernatant was discarded, then the pellet was air-dried.

#### 2.3.2. Acetone Precipitation

The four times volume (1000 µL) of 100% cold acetone was added to 250 µL of urine and mixed gently. The mixture was kept at −80 °C for 30 min, then the sample was transferred at −20 °C for overnight. The sample was centrifuged at 12,000× *g* for 40 min. The supernatant was removed, then the pellet was washed with 1000 µL of 80% cold acetone. The sample was centrifuged at 12,000× *g* for 5 min. The supernatant was discarded, then the pellet was air-dried.

#### 2.3.3. Acetonitrile Precipitation

The three times volume (750 µL) of acetonitrile was added to 250 µL of urine and mixed well for 5 min. The sample was centrifuged at 12,000× *g* for 15 min. The supernatant was discarded, then the pellet was air-dried.

#### 2.3.4. Methanol/Chloroform Precipitation

The equal volume (250 µL) of 100% methanol and 62.5 µL of chloroform were added to 250 µL of urine and mixed well for 5 min. The sample was centrifuged at 12,000× *g* for 15 min. The supernatant was removed without attaching the interface layer (protein fraction) by pipette. Then, 250 µL of 100% methanol was added to the sample and mixed gently for 5 min. The sample was centrifuged at 12,000× *g* at 25 °C for 15 min. The supernatant was discarded, then the pellet was air-dried. All the protein pellets were dissolved in 100 µL of 8 M urea/50 mM Tris-HCl (pH 8.0).

### 2.4. Protein Assays for Recovery Check

An aliquot (250 µL) of each urine sample was mixed with 250 µL of 2 M urea/100 mM Tris-HCl, 100 mM EDTA (pH 8.0) for 5 min before the measurement of protein concentration in each urine sample. To measure protein amount in the urine pellet extracted by each precipitation method, the pellet was dissolved in 250 µL of 1 M urea/50 mM Tris-HCl, 50 mM EDTA (pH 8.0) and mixed well for 5 min. These samples were used for protein assay by protein assay dye reagent concentrate (Bio-Rad, Hercules, CA, USA), according to the manual. Briefly, 40 µL of each sample was applied into three wells of 96 wells plate (triplicate), and 160 µL of 4 times diluted Bio-Rad protein assay reagent was added to each well on the plate. The plate was analyzed by Bio-Rad 3500 plate reader (Bio-Rad) at 595 nm wavelength with 405 nm for the reference. BSA solutions were prepared at concentrations of 150, 100, 75, 50, 37.5, 25, 12.5, 6.25 µg/mL with 1 M urea/50 mM Tris-HCl, 50 mM EDTA (pH 8.0) as standards. The recovery rate of precipitated urine protein was calculated by using the following formula:Protein recovery rate: R (%) = P_ppt_ (precipitated protein amount)/P_u_ (urine protein amount) × 100

### 2.5. Protein Profiles of the Pellets Examined by SDS-PAGE

The precipitated pellet was dissolved in 50 µL of 1 × SDS sample buffer (2% SDS, 62.5 mM Tris–HCl (pH 6.8), 10% glycerol, 0.01% bromophenol blue, 50 mM DTT) and boiled at 95 °C for 5 min. Then, the samples were separated by SDS-PAGE (10% gel), and the proteins were visualized by silver staining using Pierce™ Silver Stain Kit (Thermo Fisher Scientific, Waltham, MA, USA) by following the manual.

### 2.6. Peptide Preparation and Purification in Solution Digestion

Total urine protein amount used in trypsin digestion was adjusted to less than 20 µg according to the protein concentration calculated by protein assay. A substantial amount of protein containing solution was taken from the precipitated sample tube, and the sample was adjusted to a volume up to 100 µL with 8 M urea/50 mM Tris-HCl (pH 8.0) buffer. The sample was treated with 2 µL of 1 M dithiothreitol (DTT) and 8 µL of 500 mM Iodoacetamide (IAA) at RT for 1 h. Then, the sample was re-treated with 1 µL of 1 M DTT for neutralizing the remaining IAA. After the treatments, the sample was diluted with 700 µL of 50 mM Tris-HCl (pH 8.0). Then, the proteins were incubated with 1 µg of trypsin (Agilent, Santa Clara, California, USA) and activated with 50 mM Ammonium bicarbonate (ABC) (pH 7.8) at 37 °C for 16 h with shaking. After the protein digestion, trypsin reaction was terminated with 1 µL of 50% trifluoroacetic acid (TFA).

### 2.7. Peptide Purification and Quantification

The digested protein sample was purified by C18 column (GL Science, Tokyo, Japan), according to the manual. Briefly, C18 column was activated by 100% acetonitrile and stabilized by 50% acetonitrile and 0.2% formic acid (FA). Then, the sample was loaded into the column and centrifuged at 3000× *g* for 90 s. Then, the trapped peptide sample was washed with 0.2% TFA twice. Finally, the sample was eluted with 95% acetonitrile, 5% FA. The eluted sample was dried up by VEC-260 vacuum dryer (Iwaki, Tokyo, Japan). The sample was re-suspended with 0.1% FA, and then the peptide concentration was quantified by Nanodrop 1000 (Thermo Fisher Scientific, Waltham, MA, USA). The sample was stored at −80 °C until analysis.

### 2.8. LC-MS/MS

Mass spectrometric analysis was performed by using QExactive plus (Thermo Fisher Scientific) online coupled with a nanoflow high-performance liquid chromatography (HPLC) system (Thermo Fisher Scientific) equipped with a trap column (2 cm × 75 µm Acclaim Pepmap 100 column) and a separation column (12.5 cm × 75 µm NTCC-360). Mobile phases used were: solution A, 0.1% FA; B, 0.1% FA, 99.9% acetonitrile. After purification, total 500 ng of tryptic peptides were injected and eluted from analytical column at a flow rate of 300 NL/min in a linear gradient of 2% B to 35% B in 120 min. The mass spectrometer was operated in a positive mode in the scan range MS and MS/MS of 350–1800 m/z and 200–2000 m/z, respectively. The 15 most intense peaks with charge state ≥2 were selected from each survey scan in data-dependent mode and subjected to Collision Induced Dissociation fragmentation. For the MS and MS/MS, scan parameters were as below; resolution, 70,000 and 17,500; AGC target, 1e6 and 5e4, respectively. The other parameter settings were as follows: collision energy, 35%; electrospray voltage, 2.0 kV; capillary temperature, 250 °C; isolation windows, 4 m/z.

### 2.9. Analysis of Proteomic Data

All MS and MS/MS data were analyzed using Proteome Discoverer 2.1 (Thermo Fisher Scientific) for protein and peptide identification. The data were queried against a Uniprot/SWISS-PROT database (v2015-08; Homo sapiens 20,203 sequences). All database search was performed using a precursor mass tolerance of ±10 ppm, fragment ion mass tolerance of ±0.02 Da, enzyme name set to trypsin, and a maximum missed cleavages value of 2. For the in-solution digestion procedure, the fixed modification was specified as carbamidomethylation of Cys. The false discovery rate (FDR) was kept at 1% at the peptide level. The emPAI value of identified proteins was used for label-free quantification. The dataset was compared by Venny 2.1 (BioinfoGP, http://bioinfogp.cnb.csic.es/tools/venny) in each method. All MS files (.raw) are accessible from the JPOST repository at URL: http://jpost.org/.

### 2.10. Additional Application

After establishing a basic workflow for urine protein preparation, it was modified with further optimization. Briefly, a 1/20 volume (12.5 μL) of 1 M Tris-HCl (pH 8.0) was added into 250 µL of urine, and then the urine was used for urine protein precipitation by M/C method. Precipitated urine proteins were treated with 100 µL of 8 M urea/50 mM Tris-HCl, 50 mM EDTA (pH 8.0) for dissolving. The urine protein samples were used for trypsin digestion, according to the method described in Section 2.6 and Section 2.7.

### 2.11. Statistical Analysis

The significance of each data was evaluated by the unpaired *t*-test. The values of *p* < 0.05 and 0.01 were judged to be significant.

## 3. Results

### 3.1. Methanol/Chloroform Precipitation is the Best Method for Urine Protein Extraction

We investigated the recovery rate and condition of precipitated urine protein to evaluate the ability of the four different kinds of precipitation method by following a study design described in Section 2.2 (Figure 1A). Because of our experience and previous reports, the recovery rate of the precipitated urine protein was one of the important parameters in LC-MS/MS analysis to achieve high accuracy and sensitivity. It is also an indispensable point in the identification not only for dominant proteins but also minor ones in the samples. Precipitated urine proteins contained contamination of colored substances, especially almost all the samples prepared by ethanol and acetone precipitation (Figure 1B). The contaminants were suspected as, mainly, urobilin, which regularly exists in the urine. Different from those methods, the samples prepared by M/C precipitation contained much small amount of the contaminants than others (Figure 1B). We checked the precipitated protein recovery rate in the samples prepared by each precipitation. We confirmed that three methods showed a high recovery rate (around 80% or more) in the average of 10 samples except for acetonitrile (ACN) precipitation (Figure 1C, Appendix A). Ethanol precipitation showed the highest value in the average of 10 samples (85%), and the maximum value was over 90%. Acetone and M/C precipitation showed 78.5% and 78.1% of the urine protein recovery rate on average, respectively. While ACN precipitation showed the lowest protein recovery rate on average (54.6%) compared with the others. The size of precipitated urine protein pellets had consistency with the value of the recovery rate in each precipitation method (Figure 1B,C and Appendix A).

The absolute amount of precipitated urine protein was investigated by silver staining (Figure 1D). In the stained gel image, the amount of precipitated urine proteins showed consistency with the value of the recovery rate. The protein band patterns and intensities were almost identical in the three precipitations (ethanol, acetone, and M/C), which showed high protein recovery rate, except ACN precipitation (Figure 1D, lanes 2,3,5). The supernatants did not contain visible remaining proteins at all in these precipitations (Figure 1D, lanes 6,7,9). The urine proteins extracted by ACN precipitation showed less amount of protein in the gel image than the other precipitations (Figure 1D, lane 4), and proteins, sized at around 60 to 100, were retained in the supernatant (Figure 1D, lane 8).

Thus, we concluded that ethanol, acetone, and M/C precipitation could extract abundant proteins from frozen urine.

### 3.2. The Urine Protein Sample Prepared by Methanol/Chloroform Precipitation Provides the Largest Number of Identified Proteins with High Specificity

We evaluated a difference in the number of identified proteins prepared by four different kinds of precipitation by LC-MS/MS analysis (Figure 2A, Appendix A). The samples prepared by ethanol and M/C precipitation showed 1039 and 1045 identified proteins in the average of 10 samples, respectively. Acetone and ACN precipitation showed a significantly small number of identified proteins than other two precipitations. These results had consistency with the number of identified peptides in each method (Figure 2B, Appendix A). Between ethanol and M/C precipitation methods, there was no significant difference in the number of identified proteins and peptides on average (Figure 2A,B). In fact, almost all sample’s analytical values were crossed in these two precipitations (Appendix A). However, 90% of samples (9–10) in M/C precipitation showed higher values than ethanol precipitation (Appendix A).

We investigated the shared and unique proteins in the samples prepared by each method (Figure 2C). In the percentage of identified proteins, 47.6% of proteins were shared throughout four methods. The comparison between two different methods, ethanol and M/C precipitation, showed the highest percentage of shared protein as 68.4% compared to the comparison of other methods. M/C precipitation showed 9.9% of the unique proteins, and it was the highest value throughout all of the four methods. Following this result, ethanol precipitation showed 7.1% of unique proteins.

Thus, a urine protein prepared by M/C precipitation has the best condition to obtain an excellent result in the protein identification by LC-MS/MS-based urine proteomics.

### 3.3. Additional Treatments Improve the Performance of Protein Identification

Although M/C precipitation showed a good performance in urine proteomics as a base method in our study, the two critical problems remained and were required to be solved in the procedure. The first point was the contamination of colored substances in the precipitated urine protein (described in Section 3.1). Some of the samples contained the contaminants, thought to be mainly urobilin, in the precipitated urine proteins. It was significant in the precipitated protein from the urine with a strong yellow color (Figure 1B, Sample No. 3,6,10; Figure 3A, left). Even after purification using a C18 column and washing with 0.2% of TFA during peptide purification, the contamination was not able to be eliminated from the peptide (Unpublished data). Actually, the contamination might affect several procedures; however, the most critical influence was on the LC-MS/MS analysis. We experienced that the contamination induced serious damage in the mass spectrometer, and the analysis itself became unstable by the interference of the ionization and the separation of ions (unpublished data). Another point was a low solubility of the precipitated urine proteins. The urine proteins extracted by organic solvent precipitation were frequently dehydrated due to which the pellet was hard to be dissolved in 8 M urea/50 mM Tris-HCl (pH 8.0), which is generally used in proteomics [11]. In this condition, a vague white layer, suspected as undissolved proteins, remained in the bottom of the tube (Figure 3B, left). The sample still contained the undissolved fraction after trypsin digestion (Figure 3C, left). These samples frequently showed abnormal TIC patters in LC-MS/MS suspected by the stacking of flow path and ionization failure (Appendix A). In these cases, the identified number of proteins were much less compared to the best performance analysis or the analysis itself failed.

To overcome these problems in urine protein preparation, we modified the M/C precipitation, as a base method, with specific applications against potential problems. We applied several chemicals or buffers in each to eliminate the contamination in the precipitated proteins. Eventually, we found that addition of 1M Tris-HCl (pH 8.0) greatly improved the proteins’ quality on M/C precipitation (Figure 3A). The precipitated urine proteins were obviously cleaned up by this treatment. The treatment did not affect the recovery rate on M/C precipitation (Unpublished data). To improve the dissolution of the precipitated urine proteins, we adopted 8 M urea/50 mM Tris-HCl, 50 mM EDTA (pH 8.0) buffer. This buffer contributed to complete dissolving of the protein compared with the general buffers without EDTA (Figure 3B). Furthermore, the samples dissolved in the buffer did not show any white remaining part after trypsin digestion (Figure 3C). Improvement in protein dissolving showed a great benefit in the performance of LC-MS/MS analysis, and thus the identification number of proteins were significantly increased in 8 M urea/50 mM Tris-HCl, 50 mM EDTA (pH 8.0) treated samples compared with the samples treated with 8 M urea/50 mM Tris-HCl. The samples treated with 8 M urea/50 mM Tris-HCl (pH 8.0) and 8 M urea/50 mM Tris-HCl, 50 mM EDTA (pH 8.0) buffers showed 935 and 992 identified proteins in the average of three samples, respectively (Figure 3D, Appendix A). The sample treated with 1 M urea/50 mM Tris-HCl, 50 mM EDTA (pH 8.0) buffer, which failed to dissolve a part of the precipitated proteins, showed the lowest number of identified proteins, 873. These results had consistency in the number of the identified peptide (Figure 3E, Appendix A). In addition to the result of protein identification, the numbers of missed cleavage sites were significantly decreased in the well-dissolved samples treated with 8 M urea/50 mM Tris-HCl, 50 mM EDTA (pH 8.0) buffer compared with other samples (Figure 3F, Appendix A). The percentages of the peptide without missed cleavage sites in the samples reached nearly 80%.

Taken together, these additional treatments with Tris-HCl and EDTA containing buffer contribute to improving the sample condition and the analysis performance in LC-MS/MS.

### 3.4. Proposal of a Workflow for Urine Proteomics

Throughout this study, we established an optimized workflow for sample preparation in LC-MS/MS-based urine proteomics (Figure 4A). In the workflow, a total of six checkpoints were established, and these should be cleared to maintain the ideal sample condition for LC-MS/MS analysis with high performance (Figure 4B). The precipitated urine protein recovery is calculated by the following formula: (Protein recovery rate: R (%) = P_ppt_ (precipitated protein amount)/P_u_ (urine protein amount) × 100) (✔1). From the results of this study, we concluded that the samples achieved high protein recovery rate (approximately 80% or more) and showed satisfactory results in LC-MS/MS, which means the number of identified proteins was reached to 1000. Dissolving condition of the precipitated urine protein must be checked as well (✔2) because the incomplete dissolving affected the trypsin digestion (Figure 3B,C). After trypsin digestion, the sample condition must be re-checked to ensure any remaining indissoluble peptide or protein (✔3).

The sample color must be considered as well (✔1, 2, 3, 4). A precipitated urine protein with brown color might have serious contamination. It was difficult to be eliminated except in precipitation step in our result; therefore, we encourage to add 1M Tris-HCl (pH 8.0) into the original urine sample (Figure 3A). M/C precipitation has strong ability to eliminate the contaminants, so the workflow concentrates to removing the contaminants in the precipitation step.

The peptide quantification is strongly recommended to adjust the suitable concentration (✔5). In the general mass spectrometry, the injection volume of the sample is limited, so to adjust the sample concentration is indispensable step to inject the suitable amount of the peptide within the acceptable sample volume. The workflow recommends adjusting the peptide concentration to 100 ng/µL (✔5). In the meantime, the peptide recovery rate is also possible to be calculated by using the value of the quantified peptide amount (not necessary but recommend doing so). Finally, the TIC and base peak chromatogram patterns must be checked by monitoring the fundamental analytic condition in LC-MS/MS (✔6). The samples which failed to clear these checkpoints frequently showed unsatisfactory TIC pattern in LC-MS/MS, so the performance of protein identification in the samples was poorer than the optimized sample (Appendix A).

Taken together, the workflow will provide optimized sample condition in urine protein and peptide sample as well as the best performance in LC-MS/MS analysis for urine proteomics.

## 4. Discussion

Urine proteomics is used in many kinds of biomedical researches. Especially for biomarker investigation, the proteomic data is required to be produced with high accuracy and sensitivity as it covers a wide range of the targets in LC-MS/MS. We have adopted the urine proteomics approach in urine biomarker investigation using a comparison of identified protein between healthy and disease samples. In this strategy, minor proteins must be detected as well as the dominant ones. Therefore, we strongly concentrated on the sample condition. In addition, no standard procedure has been established for the sample preparation step for LC-MS/MS-based urine proteomics. Many studies identified novel urine biomarker using a proteomic approach; however, each study used a different protocol for the sample preparation, at least in the urine protein extraction step. As a result, it was difficult to compare and to evaluate the results between the studies. This is a critical disadvantage for making a consensus in this field.

In this study, we adopted organic solvent precipitation in urine protein extraction. The methods itself have been well recognized as one of the standard methods for a long time ago, so we selected it and challenged to modify the method [12]. M/C precipitation is frequently used for protein extraction from biofluid samples, including urine. However, several variations in the ratio of methanol:chloroform exist. Hence, we defined a suitable rational balance between these organic solvents in urine protein precipitation. Our protocol achieved around 80% of precipitated protein recovery rate (Figure 1C). In addition, the number of protein identification of the sample was much better than other methods. M/C precipitation showed a strong ability to eliminate contamination in the precipitated proteins, and Tris-HCl addition enhanced the ability; thus, the condition of precipitated urine proteins was cleaned up as much as possible. Although the exact mechanism of the effect is still unknown, the procedure itself is available with high stability. We tried other methods, such as dialysis and filtration, to remove contaminants from urine. However, both of those methods didn’t work well. Upon dialysis, we found critical sample loss from original urine, so the precipitated protein recovery rate was decreased. Filtration also didn’t work well because the filer was immediately stacked by colored contaminants. Hence, we emphasize that the combination of Tris-HCl pre-treatment and M/C precipitation is the most effective way for urine protein extraction.

We were really interested in the specific effect of EDTA in protein dissolving. Urine regularly contains much amount of salts (eventually these salts produce positively charged ions in the solution), and the salts might interfere with protein dissolving. EDTA is a well-known chelator for positively charged ions, such as Ca^2+^ and Mg^2+^; therefore, the beneficial effect of the chemical in protein dissolving might be natural. Furthermore, EDTA contributes to trypsin digestion because Ca^2+^ inhibits trypsin activity. It seems the reason why the sample treated with EDTA containing buffer reduced missed cleavages. While we must notice that a high concentration of urea is still required for enough dissolving of urine proteins. Throughout the study, we confirmed that the sample prepared by following the workflow showed good performance because the results of protein identification provide reliable proof. Total 150 urine samples originated from healthy volunteers were analyzed following the workflow, then these samples showed over 1000 identified proteins on average (unpublished data). We analyzed the samples using three different mass spectrometers (QExextive plus, Orbitrap fusion (Thermo Fisher Scientific), and Triple TOF 5600 (SCIEX, Framingham, MA, USA)). We experienced that all machines showed a similar trend as analysis using the sample prepared by the workflow identified the largest number of proteins. It has been considered that the workflow provides much stable analytic condition than ever before in urine proteomics.

Another thing we must consider is the analysis of a patient’s urine sample using our workflow. Patients with some diseases showed severe protein urea; therefore, the workflow must be ensuring such a high amount of urine protein condition. Actually, we tried to adopt the workflow for the analysis of diabetic urine. We found an optional pre-treatment in which the patient’s urine must be diluted to adjust to a suitable range of urine protein. Otherwise, the urine protein may be precipitated only for the dominant one, such as albumin. The patient’s urine containing protein as almost the same amount as healthy urine can be processed by M/C normally, and subsequently by trypsin digestion and peptide purification. Throughout the peptide preparation to LC-MS/MS analysis, the protein and peptide amount is exactly determined in our workflow, and thus, the initial urine protein amount in the sample is no longer considered.

In addition, a standard control of healthy urine must be set for the analysis of patient’s urine. It is hard to gain healthy urine from the patient before he/she was diagnosed. A standard database of healthy urine must be established, so we are going to create the database with our workflow.

This workflow contributes to establishing a standard procedure in urine proteomics. It can be widely used in urine proteomics not only for diagnosis using urine biomarker but also basic biomedical research, such as physiology and pharmacology. In addition, it might contribute to the establishment of urine proteomic database when the workflow would be used by each study in the future.

## Figures and Tables

**Figure 1 mps-02-00046-f001:**
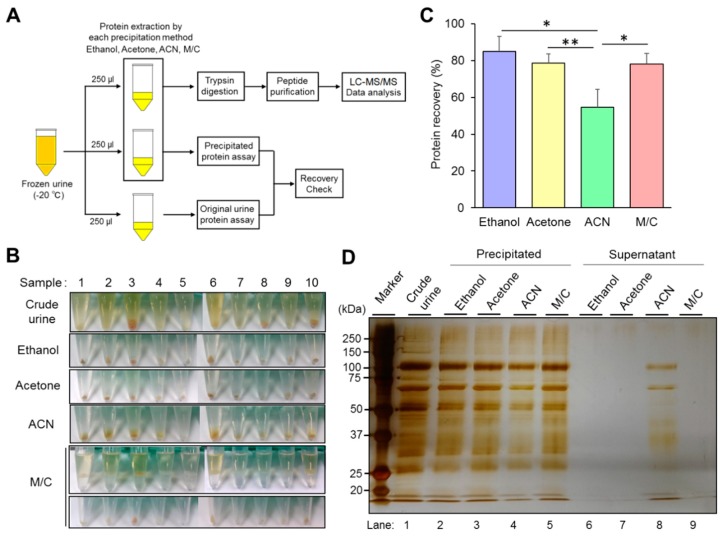
Characterization of each precipitation method. (**A**) The design of study for the characterization and evaluation of the performance in each precipitation. Crude urine was stored at –20 °C, then thawed at 37 °C before use. The 250 μL aliquots of urine samples were used for the investigation of precipitated urine protein recovery rate and protein identification, respectively. (**B**) The condition of precipitated urine protein in each precipitation method. The pictures indicate the condition of final precipitated urine proteins prepared by ethanol, acetone, acetonitrile (ACN), and methanol/chloroform (M/C) precipitation. Only for M/C precipitation method, the pictures indicated the condition both after initial centrifugation and final precipitated proteins. (**C**) The recovery rate of precipitated urine protein in each method. The urine protein concentrations in crude and precipitated samples were measured by Bradford protein assay. The recovery rate of precipitated protein was calculated by following the formula indicated in materials and methods in Section 2.4. (**D**) Silver staining image of the proteins in precipitated and in supernatant prepared by each precipitation. The proteins in those samples were separated by SDS-PAGE and were visualized by silver staining. The total protein amount loaded on the gel was adjusted with an identical amount originated from 20 μL of crude urine. (**B**,**C**) The data are based on 10 urine samples originated from different health volunteers. (**D**) The data is representative of at least three independent experiments. The Student’s *t*-test was used to analyze data for significant differences. Values of * *p* < 0.05, ** *p* < 0.01 were regarded as significant.

**Figure 2 mps-02-00046-f002:**
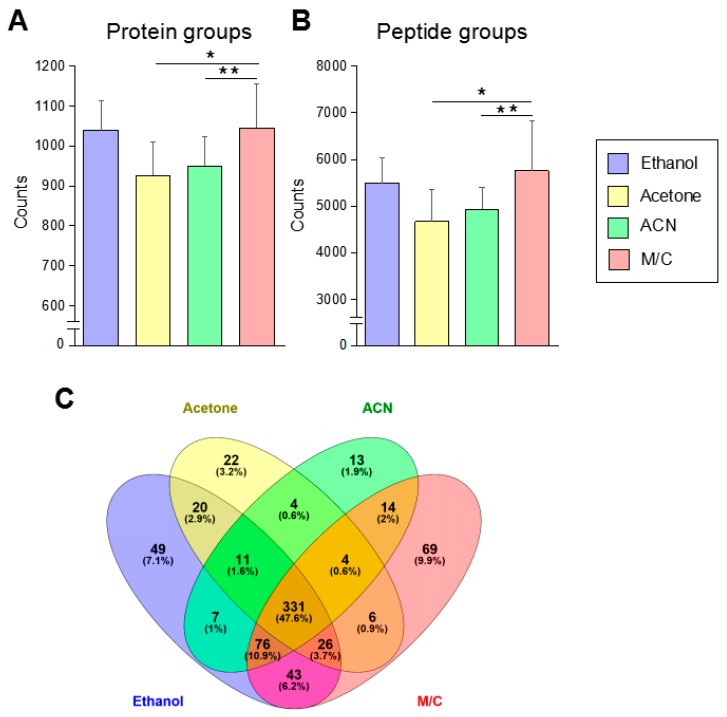
The protein and peptide identification in each precipitation method. (**A**,**B**) The identification number of protein and peptide. The samples prepared by each precipitation method were analyzed by LC-MS/MS. The proteomic data were analyzed by Proteome discoverer 2.1. Each data was originated from the result in the average of 10 samples. (**C**) The shared and unique protein in each precipitation method. Independent proteomic data provided from each precipitation were analyzed for the isolation of common or unique protein in each group (n = 10). These data were analyzed by Venny 2.1 to make a Venn diagram to indicate shred protein and unique protein between four different precipitation methods. The Student’s *t*-test was used to analyze data for significant differences. Values of * *p* < 0.05, ** *p* < 0.01 were regarded as significant. ACN: acetonitrile; M/C: methanol/chloroform.

**Figure 3 mps-02-00046-f003:**
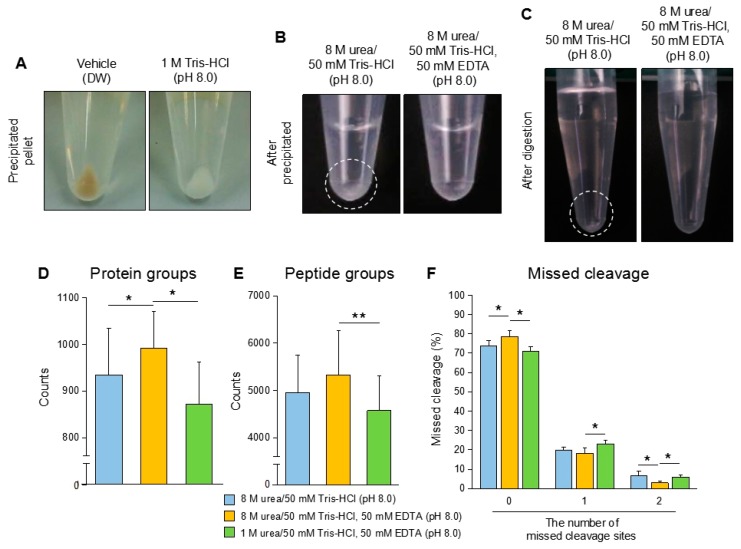
The additional applications in methanol/chloroform (M/C)-based urine protein precipitation. (**A**) Tris-HCl pre-treatment contribute to removing contaminants from precipitated urine protein. Crude urine was treated with 12.5 μL (1/20 of volume) of 1 M Tris-HCl (pH 8.0) or vehicle control (DW), then the urine proteins were extracted by M/C precipitation. (**B**) The EDTA containing buffer enhances the dissolution of precipitated urine protein. The precipitated protein samples were dissolved in 8 M urea, 50 mM Tris-HCl, 50 mM EDTA buffer (pH 8.0) or 8 M urea, 50 mM Tris-HCl (pH 8.0), respectively. (**C**) The condition of the peptide solution after digestion with trypsin. Precipitated proteins in (**B**) were treated with trypsin. (**D**–**F**) The number of identified proteins in the samples treated with each buffer. The urine proteins extracted by M/C precipitation were dissolved in 8 M urea, 50 mM Tris-HCl (pH 8.0); 8 M urea, 50 mM Tris-HCl, 50 mM EDTA (pH 8.0); 1 M urea, 50 mM Tris-HCl, 50 mM EDTA (pH 8.0), respectively. The precipitated urine samples were treated with trypsin. The samples were analyzed by LC-MS/MS, and proteomic data were analyzed by Proteome discoverer 2.1. (A–C) Data are representative of at least three independent experiments. (D–F) The data are based on three urine samples originated from different health volunteers in each buffer condition. Student’s *t*-test was used to analyze data for significant differences. Values of **p* < 0.05 and ***p* < 0.01 were regarded as significant.

**Figure 4 mps-02-00046-f004:**
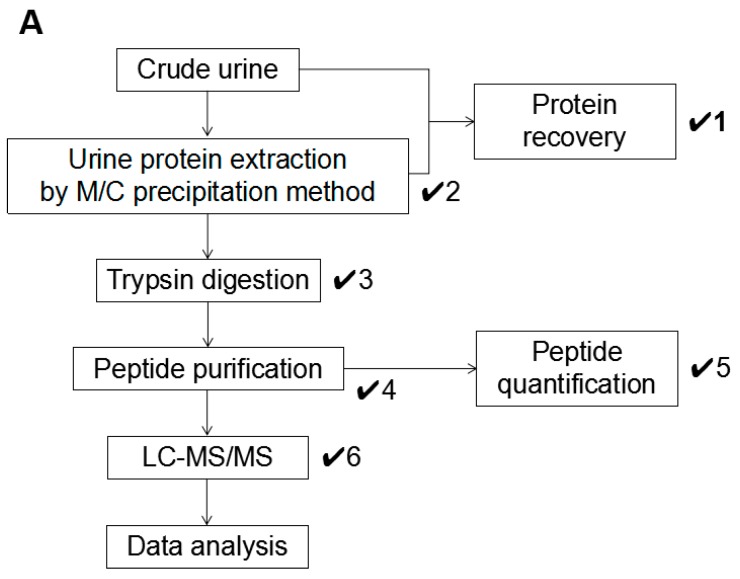
An optimized workflow for urine proteomics. (**A**) A workflow of urine protein preparation in LC-MS/MS analysis. Total of six checkpoints was established as indicated “✔.” (**B**) Summary of each checkpoint. Evaluation standards in each checkpoint were listed.

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
