# Peer review of "The Optimized Workflow for Sample Preparation in LC-MS/MS-Based Urine Proteomics"

_mps, 2019, doi:10.3390/mps2020046_

Round 1

Reviewer 1 Report

This is a good work that needs to be shared among the readers but the authors miss the most significant point while working on these experiments and coming with conclusions. The fundamental weakness of this work is to not recognize the dependence of precipitation of proteins on the chemical composition and pH of the urine. Unlike blood, urine samples are very heterogenous in their chemical composition and pH. Chemical precipitation being heavily dependent on these factors, it is very risky to propose precipitation of urine proteins with different  solvents. One such work has been published in 2008 (PMID 18459997) where precipitation method was compared with dialysis followed by lyophilization, and centrifugal filtration (that uses a set size to filter low molecular weight nonproteinous cmponents and enriches for protein without having any confounding effect of the chemical composition of the pH. I recommend that the authors to compare and comment on the two other methods. This will be good for the community and those who are thinking of using urine proteomics work. 

Author Response

Dear reviewer 1

We appreciate your comment. Your opinions and suggestions are just our concern. We seriously discussed the points following your suggestion, and we could reply as written below.

Comment) This is a good work that needs to be shared among the readers but the authors miss the most significant point while working on these experiments and coming with conclusions. The fundamental weakness of this work is to not recognize the dependence of precipitation of proteins on the chemical composition and pH of the urine. Unlike blood, urine samples are very heterogenous in their chemical composition and pH. Chemical precipitation being heavily dependent on these factors, it is very risky to propose precipitation of urine proteins with different solvents. One such work has been published in 2008 (PMID 18459997) where precipitation method was compared with dialysis followed by lyophilization, and centrifugal filtration (that uses a set size to filter low molecular weight nonproteinous cmponents and enriches for protein without having any confounding effect of the chemical composition of the pH. I recommend that the authors to compare and comment on the two other methods. This will be good for the community and those who are thinking of using urine proteomics work.

A) Urine is heterogeneous sample, so we must keep stable sample condition with additional treatment. In fact, we firstly considered urine pH. Most of urine pH were slightly shifted to acidic (pH5-6) during storage at -20. In this situation, the thawed urine sample contained abundant protein aggregation, and direct using the sample for protein precipitation leaded to increasing of contamination of colored substance. We suspected that low pH towered to generating protein aggregation by ion balance, so that contaminants were trapped into the aggregation and it be difficult to be removed during precipitation step even using M/C precipitation which basically have strong ability to clean up contaminants from precipitated proteins.

About removing of the colored contaminants, we tried dialysis and column filtration. Dialysis was at least working well for color removing from precipitated proteins. However, this treatment was a serious cause of sample loss means which the sample treated by dialysis frequently showed a low recovery rate of the precipitated protein.We guessed small proteins and peptides were elucidated during dialysis because the pore size and dialysis buffer were difficult to be optimized for each urine sample. Filtration method was basically not working well. When we challenged to clean up the contaminants in urine as a pre-treatment, the filter surface was covered by brown or yellow contaminants immediately. Eventually, the column was stacked. No other treatment archived to removed colored contaminants except Tris-HCl addition and M/C precipitation combination in our result.

We had to explain more detail about our challenge and experience in urine pre-treatment prior protein precipitation. These points are newly added in discussion.

I wish these responses are convince you.

Best regards,

Reviewer 2 Report

This research was aimed to optimize sample preparation for urine proteomics. It was focused on comparison of three precipitation methods. Although the data are solid and convincing. More information should be provided before it can be published. My comments are listed as follows.

1.      The authors should describe how they handled urine samples more clearly. In general, protein samples should be stored in an ultra-low temperature freezer (-80 °C). However, the urine samples in this study were stored at a higher temperature ( -20°C). If storing time is too long, the urine proteins are easy to be degraded by enzymes. By the way, long storage at a low temperature can also cause loss of protein due to protein precipitation. The authors should mention their processing condition in the manuscript.

2.      As mentioned in the manuscript, some yellow contaminants were precipitated together with the urine proteins. Theoretically, if these uncertain substances are small enough, it is feasible to remove them by dialysis or ultrafiltration. This point should be discussed in the article.

3.      Study of human samples should be allowed by IRB. Did this study be approved by IRB?

4.      In addition, problems that may be encountered for analyzing patients’ samples should be pointed out. For example, urine from patients with proteinuria usually have a high protein concentration. Frequency of collision of protein molecules become high when protein concentration is elevated, leading to easier precipitation of protein. That may cause a bias in terms of comparison of absolute levels of protein between patients and healthy subjects. The authors should discuss how to avoid this problem, especially when a label-free proteomics is performing.

5.      In terms of clinical study, an absolute concentration of a protein is more important than a relative level of a protein between health and disease. How to calculate the original concentration by using proteomics should also be discussed, even if it is not possible by only using MS analysis.

6.      Since loss of proteins is not avoidable and the recovery rate of protein is not 100% by using precipitation methods, the relative levels of protein might be different between the original urine solution and the final protein precipitate. For example, a protein with an original level 30% and a recovery rate 60% will give a final level 18%, while a protein with an original level 60% and a recovery rate 30% will also give a final level 18%. As a result, different original levels become a same level after precipitation. It seems that recovery rate for each individual protein is hard to estimate. The author should mention this issue in the manuscript.

Author Response

Dear reviewer 2

     I appreciate your deep reading and well understanding in our manuscript. Your comments are totally just in the focus of our study, and its are great help to improve our study so much. I would like to reply your comments.

1. The authors should describe how they handled urine samples more clearly. In general, protein samples should be stored in an ultra-low temperature freezer (-80 °C). However, the urine samples in this study were stored at a higher temperature ( -20°C). If storing time is too long, the urine proteins are easy to be degraded by enzymes. By the way, long storage at a low temperature can also cause loss of protein due to protein precipitation. The authors should mention their processing condition in the manuscript.

A) I agree with your point which urine storage method influence protein stability. It is a serious problem, so we normally stored the urine at -20℃ upto 6 month. We found that the urine stored at -80℃ tended to generate more stained contaminants in the precipitated protein than the urine stored at -20℃. From the reason, we stored the urine samples at -20℃, and used in short term as much as possible. We had to show the exact handling of urine samples, so we modified material and methods for this point.      

2. As mentioned in the manuscript, some yellow contaminants were precipitated together with the urine proteins. Theoretically, if these uncertain substances are small enough, it is feasible to remove them by dialysis or ultrafiltration. This point should be discussed in the article.

A) We tried dialysis and filtration to remove contaminants from urine sample. Dialysis might be working well because most of the color was removed in the precipitated protein from dialyzed urine. However, the precipitated protein recovery in the dialyzed urine was less than the sample from non-dialyzed urine. We guessed critical protein or peptide loss during dialysis. Actually, it was hard to select suitable pore size of dialysis membrane or composition of dialysis buffer, because urine samples were generally heterogeneous. Filtration didn’t work well by critical stacking of the filter with contaminants in the urine sample. When we loaded urine sample for filtration, the filter was immediately stained by colored contaminants, and the urine proteins weren’t eluted by stacking of  the filter.

Hence, we emphasized the way to remove the contaminants by using the combination of pre-treatment with Tris-HCl and M/C precipitation. This is the best way to remove contaminants from precipitate urine protein as much as possible without critical sample loss.

We had better to write these detail background. We modified discussion with these points..  

3. Study of human samples should be allowed by IRB. Did this study be approved by IRB?

A) The this study has been approved by IRB in Niigata University. So we added about IRB at Material and Methods section.

4. In addition, problems that may be encountered for analyzing patients’ samples should be pointed out. For example, urine from patients with proteinuria usually have a high protein concentration. Frequency of collision of protein molecules become high when protein concentration is elevated, leading to easier precipitation of protein. That may cause a bias in terms of comparison of absolute levels of protein between patients and healthy subjects. The authors should discuss how to avoid this problem, especially when a label-free proteomics is performing.

A) I totally agree this point. As you mentioned, patients’ sample, such as diabetes, frequently showed high concentration of urine protein. This point cannot be ignored in sample preparation for urine proteomics.

Actually, we tired to use our workflow for diabetic urine, then we figured out the necessary modification in the sample preparation for patient urine with extreamly high amount of protein. This kind of urine must be diluted (reduce the initial volume), so that our workflow can precipitate the urine protein with high recovery rate, and the sample is analyzed with the same performance as healthy urine. If we used the urine without dilution, the recovery rate is probably reduce, and the character of precipitated protein may be not uniform.

In the trypsin digestion, we used exact same amount of protein throughout the samples, therefore no influence was observed even in the patient urine sample. We also used the same amount of peptide for LC-MS/MS analysis. Thus, the ionization or collision aren’t influenced by initial protein amount even the sample is originated from diabetic.

From these approaches, the differences of initial protein amount in each urine isn’t to be a bias in each analysis. Initial difference of protein amount between the samples are canceled after M/C precipitation step. The subsequent step should be performed with same procedure for all of samples.

We had to write comments about these points. The discussion was modified.      

5. In terms of clinical study, an absolute concentration of a protein is more important than a relative level of a protein between health and disease. How to calculate the original concentration by using proteomics should also be discussed, even if it is not possible by only using MS analysis.

A) In our workflow, the original urine protein concentration is measured by general protein assay, not by MS. We have already optimized the method for urine protein assay with special buffer and proper standard solution. The EDTA containing buffer (written in materials and methods) achieved to dissolve protein aggregation, which was frequently contained in frozen urine sample, so that the measurement itself was very stable and reliable. In addition, the original urine color which affects absorbance can be ignored by dilution in the protein assay

In the clinical study, as you mentioned, it is hard to correct healthy urine sample from the same person who is disease currently. We are going to establish the average value (range) of urine protein in healthy persons using the urine from healthy volunteers. Once, the value was established, we can distinguish the protein amount in each urine sample is normal or abnormal. Eventually, the classification will be useful in patient urine processing and analysis which we just mentioned in “comment 4”.   

6. Since loss of proteins is not avoidable and the recovery rate of protein is not 100% by using precipitation methods, the relative levels of protein might be different between the original urine solution and the final protein precipitate. For example, a protein with an original level 30% and a recovery rate 60% will give a final level 18%, while a protein with an original level 60% and a recovery rate 30% will also give a final level 18%. As a result, different original levels become a same level after precipitation. It seems that recovery rate for each individual protein is hard to estimate. The author should mention this issue in the manuscript.

A) Yes, we had a same concern when we started this study. As you mentioned, each urine sample contains different concentration and % of protein. Therefore, we should not use the value for protein recovery calculation. One of the our proposal to measure exact urine protein recovery rate is to use the value of “absolute amount of protein”. In protein assay step, we measured protein amount in the urine (X ug). Then, we also measured the protein amount in precipitated (Y ug). This strategy provides the calculation “Y/X x100=Z (recovery rate (%))”. The value of Z is not affected by original urine concentration (ug/mL) or ratio (%), because we used the value of original amount of urine protein (ug). We agree your opinion which urine environment is not homogenous. Hence, we can only do to using absolute urine amount for recovery calculation.

I wish these responses are convince you.

Best regards,

Round 2

Reviewer 2 Report

The authors have appropriately revised the manuscript and I feel that quality of the paper has been greatly improved.